# Transcriptomic Profiling of Cold Stress-Induced Differentially Expressed Genes in Seedling Stage of *Indica* Rice

**DOI:** 10.3390/plants12142675

**Published:** 2023-07-17

**Authors:** Tao Yan, Meng Sun, Rui Su, Xiaozhong Wang, Xuedan Lu, Yunhua Xiao, Huabing Deng, Xiong Liu, Wenbang Tang, Guilian Zhang

**Affiliations:** 1College of Agronomy, Hunan Agricultural University, Changsha 410128, China; tyan@hunau.edu.cn (T.Y.); sunmeng@163.com (M.S.); surui@163.com (R.S.); wangxiaozhong@163.com (X.W.); luxuedan1@126.com (X.L.); xyhkemy@163.com (Y.X.); denghuabing@126.com (H.D.); xiongliu@whu.edu.cn (X.L.); 2Hunan Provincial Key Laboratory of Rice and Rapeseed Breeding for Disease Resistance, Changsha 410128, China; 3Hunan Hybrid Rice Research Center, Hunan Academy of Agricultural Sciences, Changsha 410128, China; 4State Key Laboratory of Hybrid Rice, Changsha 410128, China

**Keywords:** cold stress, RNA-seq, rice, cold-transcription factors, gene ontology, KEGG pathway analysis

## Abstract

Cold stress significantly constrains the growth, development, productivity, and distribution of rice, particularly the *indica* cultivar, known for its susceptibility to cold, limiting its cultivation to specific regions. This study investigated the genes associated with cold responsiveness in the roots of two *indica* cultivars, SQSL (cold-tolerant) and XZX45 (cold-susceptible), through transcriptome dynamics analysis during the seedling stage. The analysis identified 8144 and 6427 differentially expressed genes (DEGs) in XZX45 and SQSL, respectively. Among these DEGs, 4672 (G2) were shared by both cultivars, while 3472 DEGs (G1) were specific to XZX45, and 1755 DEGs (G3) were specific to SQSL. Additionally, 572 differentially expressed transcription factors (TFs) from 48 TF families, including *WRKY*, *NAC*, *bHLH*, *ERF*, *bZIP*, *MYB*, *C2H2*, and *GRAS*, were identified. Gene Ontology (GO) enrichment analysis revealed significant enrichment of DEGs in the G3 group, particularly in the “response to cold” category, highlighting the crucial role of these specific genes in response to cold stress in SQSL. Furthermore, Kyoto Encyclopedia of Genes and Genomes (KEGG) analysis indicated pronounced enrichment of DEGs in the G3 group in metabolic pathways such as “Pyruvate metabolism”, “Glycolysis/Gluconeogenesis”, and “Starch and sucrose metabolism”, contributing to cold tolerance mechanisms in SQSL. Overall, this study provides comprehensive insights into the molecular mechanisms underlying cold responses in the *indica* cultivar, informing future genetic improvement strategies to enhance cold tolerance in susceptible *indica* rice cultivars.

## 1. Introduction

Rice (*Oryza sativa* L.) plays a critical role as a staple crop, providing more than half of the world’s population with their primary source of carbohydrates [1]. Its cultivation encompasses diverse ecological conditions, ranging from upland to deep-water environments and spanning various altitudes, climates, and soil types [2]. However, rice production faces numerous challenges due to abiotic stresses, including cold, high temperatures, drought, salinity, and flooding. Cold stress, in particular, poses a significant obstacle to rice farming, leading to chlorosis, reduced seedling growth, stunting, withering, decreased tillering, and ultimately reduced productivity in susceptible cultivars [3]. The global impact of low-temperature stress on rice is substantial, affecting approximately 15 million hectares in 24 countries. In South and Southeast Asia alone, about 7 million hectares of land are unsuitable for rice cultivation due to the detrimental effects of cold stress [4]. Unanticipated cold weather events, resulting from extreme climatic conditions, further compound this challenge [5]. Cold stress has a significant impact on rice yield in countries such as Australia, Korea, China, Japan, and India, where dry-season boro rice is particularly vulnerable during the seedling stage, affecting approximately 4 million hectares of rice-growing areas and causing growth delays. Moreover, these delays coincide with high-temperature stress during the flowering stage, leading to diminished yields [6]. Consequently, the development of high-yielding rice varieties with cold stress tolerance assumes paramount importance in enhancing rice production in these vulnerable regions. Additionally, a comprehensive understanding of the molecular mechanisms underlying chilling stress tolerance is of critical significance.

Plants have evolved diverse mechanisms to counteract stresses, adapting to different stress types, intensities, and ecological contexts [7]. Various stressors such as wounds, pathogen invasion, UV radiation, flooding, drought, salinity, heat, and cold stress, either individually or in combination, trigger an increase in reactive oxygen species (ROS) levels [8]. Maintaining ROS homeostasis assumes a crucial role in regulating the expression of cold-responsive genes in rice [9]. Previous investigations have revealed the involvement of cold stress binding factors (*CBFs*) in facilitating ROS detoxification [10]. Furthermore, a ROS-mediated regulatory module has been identified as an early component of the chilling stress response pathway in rice [11]. Reactive oxygen species are considered a converging point for multiple gene networks in response to stress [8]. Transcription factors (TFs) serve as key regulators, orchestrating the expression of downstream target genes within a network. Prior research has underscored the significance of *WRKY*, *MYB*, *bHLH*, *bZIP*, *AREB*/*ABF*, *NAC*, and *DREB1*/*CBF* in governing the expression of genes responsive to salinity, temperature, and drought stress in rice [12]. Heat shock factor (*HSFs*) genes have also emerged as crucial nodes for cross-talk in the rice stress response [13]. Recent discoveries have demonstrated that a common set of TFs, coupled with stress signaling networks and upstream regulatory gene regulons, govern plant responses to multiple stresses [10,14,15,16]. Target TFs operate in a network mode to regulate molecular, biochemical, genetic, and physiological processes in response to various stressors [17]. The binding of TFs in conjunction with cis-elements provides a structural basis for generating distinct patterns of gene expression [18]. In a recent study involving the cloning of the *COLD1* quantitative trait locus, a mechanism underlying chilling tolerance was elucidated, whereby the QTL interacts with the alpha subunit of G-protein to activate the Ca^2+^ channel. This activation enhances the GTPase activity of G-protein, ultimately leading to the expression of chilling tolerance in *japonica* rice [19].

The transcriptome encompasses the entirety of transcripts present within a cell, encompassing their types and quantities, during a specific developmental stage or physiological condition [20]. With the advent of next-generation sequencing (NGS), RNA sequencing (RNA-seq) has emerged as an innovative technology for transcriptome analysis, enabling precise identification of differentially expressed genes (DEGs) and potential molecular mechanisms [20,21,22]. RNA-seq primarily relies on NGS platforms that encompass comprehensive systems for library construction, sequencing, and analysis [22]. Over the past decade, RNA-seq has been extensively employed to compare the transcriptomes of cold-tolerant and cold-sensitive plants under low-temperature stress conditions [23,24,25,26,27].

In this study, we conducted an investigation into the transcriptomic dynamics of the root tissues in two *indica* rice cultivars, SQSL and XZX45, under conditions of optimal temperature and cold stress using high-throughput RNA-seq technology. The aim was to gain insights into the intricate molecular responses occurring at the transcriptome level in these cultivars when subjected to cold stress. By employing a comprehensive RNA-seq approach, we sought to elucidate the complex regulatory mechanisms and identify key genes and pathways involved in the *indica* rice root’s response to cold stress.

## 2. Results

### 2.1. Phenotyping of Contrasting Genotypes for Cold Stress Tolerance and Recovery

Two cultivars, SQSL and XZX45, were subjected to a controlled low-temperature treatment at 8 °C for a duration of 5 days, followed by a subsequent 5 days recovery treatment. Under the imposed low-temperature stress conditions, the cold-susceptible genotype XZX45 exhibited notable withering in both its belowground and aboveground organs. Conversely, in comparison to XZX45, the cold-tolerant genotype SQSL displayed sustained growth and maintained normal leaf coloration, with its belowground and aboveground components demonstrating substantial recovery, approaching near-optimal levels (Figure 1A). Following the 5 days low-temperature stress period and subsequent 5 days recovery treatment, SQSL exhibited an impressive average survival rate of 99%, while the survival rate of XZX45 was a mere 8% (Figure 1B).

### 2.2. Transcriptome Sequencing and Differentially Expressed Genes in Response to Cold Stress

We employed RNA-seq technology to investigate the transcriptome dynamics in the roots of SQSL and XZX45 under optimal temperature and cold stress conditions. A total of 36 samples, including three independent biological replicates at three different time points, namely 1, 3, and 5 days after cold treatment, were analyzed, resulting in 241.2 Gb of clean data (Appendix A). The clean reads showed high quality, with average Q20 and Q30 values exceeding 97.55% and 90.59%, respectively (Appendix A). After mapping the reads to the rice reference genome of *cv*. *Nipponbare*, an average mapped read ratio of 85.36% (ranging from 70.20% to 94.60%) was achieved (Appendix A). Read counts mapped to each gene were determined using featureCounts [28], and transcripts per million (TPM) values were calculated. Genes with a TPM ≥ 1 were considered expressed (Appendix A).

To assess the global differences in transcriptomes between SQSL and XZX45 under cold stress, Pearson correlation coefficients (PCC) were calculated based on TPM values of expressed genes for all samples from the three time points. The PCC values of the three biological replicates of SQSL and XZX45 exceeded 0.9 (Figure 2A,B). Principal component analysis (PCA) was conducted, indicating significant differences in transcriptomes among different genotypes and treatments (Figure 2C). Hierarchical cluster analysis further confirmed the distinct clustering of the three biological replicates of SQSL and XZX45, resulting in the division of the 36 samples into 12 groups, highlighting the significant differences in transcriptomes among the experimental groups (Figure 2D).

Differentially expressed genes (DEGs) under cold stress conditions were identified using a threshold of FDR < 0.05 and |log_2_foldchange| > 1. In SQSL, a total of 16,283 DEGs were identified, with 10,298 (5587 up- and 5341 down-regulated), 11,072 (4989 up- and 6083 down-regulated), and 11,554 (5134 up- and 6420 down-regulated) DEGs found in SQSL1d (SQSL after 1 day of cold treatment), SQSL3d, and SQSL5d, respectively. Meanwhile, 19,040 DEGs were identified in XZX45, with 11,178 (6000 up- and 5178 down-regulated), 14,626 (6875 up- and 7751 down-regulated), and 15,192 (7021 up- and 8171 down-regulated) differentially expressed genes in XZX451d, XZX453d, and XZX455d, respectively (Figure 3).

Comparing the number of DEGs between SQSL and XZX45, it was evident that XZX45 had notably more DEGs than SQSL. While the number of DEGs in SQSL and XZX45 after 1 day of cold treatment did not show a significant difference, substantial differences were observed after 3 and 5 days of cold treatment (Figure 3B,C). These results indicate that cold stress had a more pronounced and sustained impact on gene expression levels in XZX45 compared to SQSL, suggesting that SQSL exhibited stronger tolerance and adaptability to cold stress than XZX45. A total of 9899 genes were detected in both genotypes across the three time points, with 8144 genes in SQSL and 6427 genes in XZX45 (Figure 3B). Among these DEGs, we identified 4672 DEGs common to both genotypes (G2), with 1755 genes specifically differentially expressed in SQSL (G3) and 3472 genes specifically differentially expressed in XZX45 (G1) (Figure 3B and Appendix A). Overall, these findings provide insights into the transcriptional dynamics and differential gene expression patterns between SQSL and XZX45 under cold stress conditions, highlighting the contrasting responses and potential molecular mechanisms underlying their cold stress tolerance.

### 2.3. Role of Transcription Factors under Cold Stress

To identify cold-responsive transcription factors (TFs) in the regulatory network of cold stress, we utilized the PlantTFDB database [29], which contains information on known or annotated TF genes in the rice genome. Out of the 1,759 TF genes belonging to 56 different TF families, a total of 572 differentially expressed TFs from 48 TF families, including *WRKY*, *NAC*, *bHLH*, *ERF*, *bZIP*, *MYB*, *C2H2*, and *GRAS*, were identified (Figure 4, Appendix A). Among these TFs, 192 were present in G1, 294 in G2, and 86 in G3 (Figure 4, Appendix A).

In terms of specific TF families, we observed the following distribution. G1 contained 12 *WRKY*, 11 *NAC*, 16 *bHLH*, 13 *ERF*, 19 *bZIP*, and 16 *MYB* TFs. In G2, we identified 34 WRKY, 31 *NAC*, 22 *bHLH*, 28 *ERF*, 20 *bZIP*, and 19 *MYB* TFs. For G3, we detected 5 *WRKY*, 6 *NAC*, 9 *bHLH*, 5 *ERF*, 5 *bZIP*, and 7 *MYB* TFs (Figure 4). Notably, the *bZIP* family had the highest number of TF DEGs in G1 (19), while *WRKY* and *bHLH* families had the highest numbers in G2 (34) and G3 (9), respectively (Figure 4). Among the nine differentially expressed *bHLH* TFs in G3, five genes (*Os01g0196300*, *Os01g0865600*, *Os03g0811400*, *Os06g0164400*, and *Os06g0193400*) were exclusively induced, and four genes (Os02g0710300, *Os03g0135700*, *Os06g0184000*, and *Os09g0474100*) were repressed after 1, 3, and 5 days of cold treatment (Appendix A). Interestingly, no G2-like TFs were identified in G3, while 10 and 10 were detected in G1 and G2, respectively (Figure 4). Additionally, two *NF-YB* TFs (*Os05g0463800* and *Os01g0935200*) were specific to G3, with *Os05g0463800* being exclusively induced and *Os01g0935200* being repressed after 1, 3, and 5 days of cold treatment (Appendix A).

These findings provide insights into the differential expression patterns and specific TF families involved in the cold stress response, highlighting the potential roles of *WRKY*, *NAC*, *bHLH*, *ERF*, *bZIP*, *MYB*, *C2H2*, and *GRAS* TFs in the regulatory network underlying cold stress tolerance in rice.

### 2.4. Gene Ontology and Pathway Enrichment Analysis of DEGs

Gene Ontology (GO) enrichment analysis was conducted to evaluate the enrichment of differentially expressed genes (DEGs) in groups G1, G2, and G3, where the statistical significance threshold was set at *p.adjust* < 0.05 (Appendix A). Our analysis identified a total of 354 GO terms encompassing diverse biological processes (BP), cellular components (CC), and molecular functions (MF). Specifically, within these GO terms, 231 were associated with BP, 40 with CC, and 82 with MF. Notably, we present the top 18 GO terms for each group (Figure 5A–C).

In the G1 group, comprised of DEGs specific to the cold-susceptible genotype XZX45 (Figure 3B), the prominent GO terms included “ubiquitin-like protein transferase activity” (GO:0019787), “endosome” (GO:0005768), “glucosyltransferase activity” (GO:0046527), “hexosyltransferase activity” (GO:0016758), and “cellular response to abscisic acid stimulus” (GO:0071215) (Figure 5A, Appendix A).

The G2 group exhibited DEGs that were differentially expressed in both the cold-susceptible genotype XZX45 and the cold-tolerant genotype SQSL (Figure 3B). Noteworthy GO terms in this group encompassed “cell wall organization or biogenesis” (GO:0071554), “cytoskeleton” (GO:0005856), “mitotic cell cycle” (GO:0000278), “microtubule cytoskeleton” (GO:0015630), and “hydrolase activity, acting on glycosyl bonds” (GO:0016798) (Figure 5B, Appendix A).

Distinctly, the G3 group comprised DEGs that were specific to the tolerant genotype SQSL (Figure 3B). The prominent GO terms in this group included “response to cold” (GO:0009409), “cytoskeleton” (GO:0005856), “mitotic cell cycle” (GO:0000278), “microtubule cytoskeleton” (GO:0015630), “transmembrane signaling receptor activity” (GO:0004888), and “cell cortex” (GO:0005938) (Figure 5C, Appendix A).

To investigate the biological processes occurring during cold stress treatment, we utilized the Kyoto Encyclopedia of Genes and Genomes (KEGG) pathway database [30]. Specifically, we performed KEGG pathway analysis and compared gene clusters’ functional profiles for the differentially expressed genes (DEGs) in groups G1, G2, and G3, where the statistical significance threshold was set at *p.adjust* < 0.05 (Appendix A). The analysis revealed the involvement of 413 DEGs across 35 pathways in the G1, G2, and G3 groups (Appendix A).

In the G1 group, which consists of DEGs specific to the cold-susceptible genotype XZX45, the top KEGG terms included “Thiamine metabolism” (dosa00730), “Endocytosis” (dosa04144), “alpha-Linolenic acid metabolism” (dosa00592), “Galactose metabolism” (dosa00052), “Homologous recombination” (dosa03440), and “Ubiquinone and other terpenoid-quinone biosynthesis” (dosa00130) (Figure 5D, Appendix A).

The G2 group exhibited DEGs that were commonly differentially expressed in both the cold-susceptible genotype XZX45 and the cold-tolerant genotype SQSL. Notable KEGG terms in this group included “alpha-Linolenic acid metabolism” (dosa00592), “Phenylalanine, tyrosine and tryptophan biosynthesis” (dosa00400), “Plant hormone signal transduction” (dosa04075), “Monobactam biosynthesis” (dosa00261), “DNA replication” (dosa03030), “Glycine, serine, and threonine metabolism” (dosa00260), “Phenylpropanoid biosynthesis” (dosa00940), and “Motor proteins” (dosa04814) (Figure 5D, Appendix A).

Distinctly, the G3 group comprised DEGs specific to the cold-tolerant genotype SQSL. The prominent KEGG terms in this group included “Galactose metabolism” (dosa00052), “Motor proteins” (dosa04814), “Biosynthesis of various plant secondary metabolites” (dosa00999), “Pyruvate metabolism” (dosa00620), “Glycolysis/Gluconeogenesis” (dosa00010), “Cyanoamino acid metabolism” (dosa00460), and “Starch and sucrose metabolism” (dosa00500) (Figure 5D, Appendix A). This suggests a balance between growth and the response to cold stress in the tolerant genotype SQSL.

Interestingly, we observed that “alpha-Linolenic acid metabolism” (dosa00592) was a common pathway between the G1 and G2 groups, “Galactose metabolism” (dosa00052) was a common pathway between the G1 and G3 groups, and “Motor proteins” (dosa04814) was a common pathway between the G2 and G3 groups (Figure 5D, Appendix A).

### 2.5. Analysis of the DEGs in GO Category: “Response to Cold” (GO:0009409) in G3 Group

We have identified a total of 1755 differentially expressed genes (DEGs) specific to the cold-tolerant genotype SQSL, referred to as the G3 group (Figure 3B). Our Gene Ontology (GO) enrichment analysis revealed a significant enrichment of DEGs in the G3 group associated with the biological process “response to cold” (GO:0009409) (Figure 4C), which was not observed in the G1 and G2 groups (Figure 4A,B). Notably, we identified 27 DEGs enriched in the “response to cold” term (GO:0009409), among which 14 DEGs, including *Os12g0137500*, *Os05g0459000*, *Os06g0164400*, *Os03g0719100*, *Os02g0686100*, and *Os09g0522000*, were exclusively induced, while 13 DEGs, including *Os01g0857200*, *Os01g0249200*, *Os08g0499300*, *Os03g0452300*, *Os02g0781400*, *Os08g0151800*, *Os03g0785900*, and *Os08g0441500*, were repressed following cold stress treatment (Figure 6A, Table 1). These findings suggest that these specific DEGs associated with the “response to cold” (GO:0009409) are actively involved in the response to cold stress in the cold-tolerant genotype SQSL.

To further explore the expression patterns of these DEGs, we compared their expression levels between the cold-susceptible genotype XZX45 and the cold-tolerant genotype SQSL following 1, 3, and 5 days of cold treatment. Among these DEGs, we identified *OsTRXh1* (*Os07g0186000*) (Figure 6A,B, Table 1), a member of the thioredoxin family encoding the h-type Trx protein. Previous studies have demonstrated the involvement of h-type Trx protein-encoding genes in the cold stress response of rice seedlings [24,31]. Our results revealed that the expression of *OsTRXh1* was up-regulated consistently at 1, 3, and 5 days of cold stress treatment (Table 1). Interestingly, we observed no significant difference in expression levels between SQSL and XZX45 after 1 day of cold stress. However, at 3 and 5 days of cold treatment, the expression levels of *OsTRXh1* were significantly higher in SQSL compared to XZX45 (Figure 6B).

Additionally, we discovered that the *OsCOLD1* (*Os04g0600800*) gene, which encodes a regulator of G-protein signaling localized on the plasma membrane and endoplasmic reticulum (ER), contributes to cold tolerance in rice (Table 1) [19]. Our findings revealed that *OsCOLD1* exhibited higher expression levels in SQSL compared to XZX45 after 3 and 5 days of cold treatment (Figure 6B, Table 1). In response to cold stress, *Drought Responsive Element Binding* (*DREB*) genes are rapidly induced. *DREB* factors, which belong to the *APETALA2*/*ethylene*-*responsive factor* (*AP2*/*ERF*) transcription factor subfamily, can recognize dehydration-responsive element/C-repeat (*DRE*/*CRT*) cis-elements. This recognition triggers the expression of cold stress-related genes, thereby enhancing cold tolerance [10]. In our study, we observed up-regulation of *OsDREB1B* (*Os09g0522000*), encoding the dehydration-responsive element-binding protein 1B, which confers chilling tolerance in rice. Specifically, OsDREB1B was up-regulated during the later response phases at 3 and 5 days of cold treatment in SQSL (Table 1, Figure 6A). Furthermore, the expression level of *OsDREB1B* in SQSL was higher than that in XZX45 at these time points (Figure 6B), suggesting its potential role in conferring cold tolerance during the late response phase in rice.

We also identified the gene *OsSIZ2* (*Os03g0719100*), encoding SUMO E3-ligases, involved in stress response and stress adaptation [32]. In our study, *OsSIZ2* was up-regulated at 1, 3, and 5 days of cold treatment in SQSL. Notably, the expression level of *OsSIZ2* in SQSL was higher than that in XZX45 after 5 days of cold treatment (Table 1, Figure 6B).

Moreover, several genes, including *OsbHLH108* (*Os06g0164400*), *OsABA3* (*Os06g0670000*), *OsWRKY30* (*Os08g0499300*), *OsRAB16B* (*Os11g0454200*), and two *TAP42*-like family genes (*Os11g0141000*, *Os12g0137500*), exhibited exclusive differential expression patterns under 1, 3, and 5 days of cold stress treatment. These genes also showed differential expression levels between SQSL and XZX45 (Figure 6A,B, Table 1). These results indicate that the DEGs enriched in the “response to cold” (GO:0009409) gene ontology category are significantly involved in different gene families and function as cold-responsive genes.

### 2.6. Validation by qRT-PCR

We conducted validation of the RNA-seq data’s accuracy and reliability by assessing the expression levels of 7 randomly selected genes. The expression profiles of these genes, determined through quantitative real-time polymerase chain reaction (qRT-PCR), exhibited a high degree of similarity with the results obtained from RNA-seq analysis (Appendix A, Appendix A).

## 3. Discussion

Given the constraints imposed by low environmental temperatures, plants have evolved diverse response mechanisms to cope with cold stress. Rice, originating from tropical or subtropical regions, is particularly susceptible to cold stress, which severely limits its productivity, an important consideration for ensuring sustainable rice production [33,34,35]. Asian cultivated rice (*Oryza sativa*) has undergone domestication from its wild relatives, namely *Oryza nivara* and *O. rufipogon*. It comprises two major subspecies, *indica* (*O. sativa* ssp. *indica*) and *japonica* (*O. sativa* ssp. *japonica*) [34,35]. The temperate *japonica* cultivars, a specific type of japonica, are commonly cultivated in regions with lower average temperatures and exhibit higher tolerance to chilling stress compared to *indica* cultivars [34,35]. While there have been several studies investigating the molecular genetics underlying cold tolerance during the seedling stage between *japonica* and *indica* cultivars [23,24,25,27,36,37], limited research has focused solely on *indica* cultivars. In this study, two *indica* cultivars, SQSL, which was classified as very highly tolerant, and XZX45, which was highly susceptible to chilling stress (Figure 1), were utilized to investigate the mechanisms involved in the cold stress response of rice using RNA-seq analysis.

A total of 9899 DEGs in both the contrasting genotypes were identified, of which 8144 genes and 6427 genes were observed in XZX45 and SQSL, respectively (Figure 3), which suggests that cold stress had a more pronounced and sustained impact on gene expression levels in XZX45 than in SQSL, indicating that the tolerance and adaptability of SQSL to cold stress were stronger than those of XZX45. Previous studies on cold and salinity tolerance also revealed a relatively different number of DEGs in contrasting genotypes [23,24,25,26]. Of 9899 DEGs, we found 4672 DEGs (G2) were common to SQSL and XZX45, 1755 (G3) and 3472 genes (G1) were only differentially expressed in SQSL and XZX45, respectively, which suggested that these 1755 specific genes may be continuously leading the tolerance to cold stress in SQSL (Figure 3B).

Transcription factors (TFs) significantly regulate various aspects of plant development, including propagation, maturation, and responses to unfavorable environmental conditions such as drought, cold, salinity, and high temperature [38,39]. Numerous studies have demonstrated the crucial roles played by different types of transcription factors (TFs) in various functional pathways within plants. For instance, *WRKY* TFs play critical roles in plant responses to biotic and abiotic stresses [40]; NAC TFs are involved in controlling numerous aspects of plant development, such as embryo development, leaf senescence, lateral root formation, and response to abiotic stresses [41], *bZIP* TFs regulate processes including pathogen defense, light and stress signaling, seed maturation, and flower development [42], *bHLH* TFs have been found to be involved in phytochrome signaling activity [43], *ERF* TFs have been implicated in diverse responses to environmental stimuli [44], and *MYB* TFs have been identified as key players in the control of anthocyanin biosynthesis [45]. We have identified a total of 572 differentially expressed TFs related to 48 TF families, including *WRKY*, *NAC*, *bHLH*, *ERF*, *bZIP*, *MYB*, *C2H2*, and GRAS (Figure 4, Appendix A). Notably, the *bZIP* family had the highest number of TF DEGs in G1 (19), while the *WRKY* and *bHLH* families had the highest numbers in G2 (34 *WRKYs*) and G3 (9 *bHLHs*), respectively (Figure 4, Appendix A). These results suggest that these TFs might be involved in response to cold stress in rice.

GO enrichment analysis showed that the DEGs in G1, G2, and G3 were enriched in different GO terms, such as “ubiquitin-like protein transferase activity” (GO:0019787), “glucosyltransferase activity” (GO:0046527) in G1, “cell wall organization or biogenesis” (GO:0071554) in G3 and “response to cold” (GO:0009409) in G3 (Figure 5A, Appendix A). Interestingly, KEGG analysis showed that most of the DEGs in G3 were enriched in “Pyruvate metabolism” (dosa00620), “Glycolysis/Gluconeogenesis” (dosa00010), and “Starch and sucrose metabolism” (dosa00500), which indicated that a balance between plant development and stress response in rice might confer cold tolerance in SQSL (Figure 5B, Appendix A).

We have found several cold-response related genes in GO term “response to cold” (GO:0009409), such as *OsTRXh1* (*Os07g0186000*) [31], *OsCOLD1* (Os04g0600800) [19], *OsDREB1B* (*Os09g0522000*) [46], and *OsSIZ2* (*Os03g0719100*) [32] (Figure 5, Table 1). These genes have been reported to respond to cold stress. The gene expression level analysis showed that *OsTRXh1*, *OsDREB1B*, and *OsCOLD1* were up-regulated after cold treatment and showed higher expression levels in SQSL than that in XZX45 (Figure 6B, Table 1), which partly explained that SQSL exhibited stronger tolerance and adaptability to cold stress than XZX45.

## 4. Materials and Methods

### 4.1. Plant Materials and Cold Treatment

The cold-tolerant genotype SQSL and cold-susceptible genotype XZX45 were used in this study. Mature non-dormant seeds of the two *indica* cultivars were distilled with 2% H_2_O_2_ for 30 min and then rinsed three times with distilled water. Rice seeds were transferred to the germination box in a growth chamber for 3 days at 22 °C in the dark. The germinated seeds were transferred to 96-well plates and then grown hydroponically. The 96-well plates were placed in a plant growth chamber (14 h-light/10 h-dark conditions) with temperatures of 28 °C and 25 °C for the light and dark conditions, respectively. The three-leaf seedlings were transferred to a growth chamber and exposed at a temperature of 8 °C (treatment) or 30 °C (control) with a 12 h light:12 h dark photoperiod as the temperature treatment. Rice roots at 1, 3, and 5 days later under control and cold treatment were harvested with three technological and biological replicates and then stored at −80 °C for RNA extraction and sequencing.

In this study, we utilized the cold-tolerant genotype SQSL and the cold-susceptible genotype XZX45. To initiate the experiment, mature non-dormant seeds of the two genotypes were subjected to a 30-min treatment with 2% H_2_O_2_, followed by rinsing with distilled water three times. Subsequently, the rice seeds were transferred to a germination box within a growth chamber and kept in the dark at a temperature of 22 °C for a period of 3 days.

Upon germination, the sprouted seeds were carefully transferred to 96-well plates for hydroponic cultivation. The 96-well plates were placed in a plant growth chamber (14 h-light/10 h-dark conditions) with temperatures of 28 °C and 25 °C for the light and dark conditions, respectively. The environmental conditions were maintained consistently throughout the experiment.

After the seedlings had developed three leaves, they were transferred to a growth chamber for the subsequent temperature treatment. The temperature conditions were set at 8 °C for the cold treatment group and 30 °C for the control group, both under a 12-h light and 12-h dark photoperiod. These conditions represented the respective cold and optimal temperature regimes.

To analyze the effects of the temperature treatment on gene expression, rice roots were harvested at 1, 3, and 5 days after the initiation of the control and cold treatments. Each sampling time point was conducted with three replicates, encompassing both technological and biological replicates. The harvested root samples were immediately stored at −80 °C for subsequent RNA extraction and sequencing procedures.

### 4.2. RNA Extraction and Sequencing

To prepare the root samples for RNA extraction, fresh root samples were ground using mortars pre-cooled with liquid nitrogen. Total RNA extraction was carried out using Trizol reagents (CAT. No. 15596018, Invitrogen, State of California, USA) according to the manufacturer’s protocol. The extracted RNA samples were further purified using the Rneasy Mini Kit (CAT. No. 74104, Qiagen, Dusseldorf, Germany) to remove contaminants and impurities. The concentration and quality of the purified RNA samples were assessed using a Colibri instrument (LB 915, Titertek Berthold, Pforzheim, Germany), which measures parameters such as RNA concentration, purity, and integrity. For sequencing, a total of 36 libraries were prepared, with each library representing a separate sample or condition. The libraries were sequenced using the BGISEQ-500 sequencer, which is a high-throughput sequencing platform capable of generating large amounts of sequencing data.

### 4.3. RNA-Seq Analysis

The raw sequencing reads obtained from the BGISEQ-500 sequencer underwent data preprocessing steps. Firstly, adapter trimming and removal of low-quality reads were performed using the fastp (v0.23.2) [47]. The resulting clean reads were then aligned to the genome sequence of *Oryza sativa* ssp. *japonica* cv. *Nipponbare* (IRGSP-1.0 2022-09-01, https://rapdb.dna.affrc.go.jp) using the HISAT2 (v2.2.1) [48]. To determine the read counts mapped to each gene, the featureCounts, tool was utilized [28]. Additionally, the transcripts per million (TPM) values for each gene were calculated using in-house R scripts. Genes with a TPM value of at least 1 were considered as expressed. Differential expression analysis was performed using the DESeq2 R package (v1.40.1) [49]. The analysis aimed to identify genes showing significant differential expression under cold stress conditions. The criteria for differential expression were set as a false discovery rate (FDR) of less than 0.05 and an absolute log2foldchange greater than 1. To gain insights into the functional annotations and pathways associated with the differentially expressed genes (DEGs), Gene Ontology (GO) enrichment analysis and KEGG pathway enrichment analysis were conducted using the ClusterProfile R package (v 4.7.1) [50]. The enrichment analysis was implemented with the false discovery rate (FDR) correction. To identify transcription factors (TFs) among the DEGs, the PlantTFDB database (http://planttfdb.gao-lab.org) [29] was utilized.

### 4.4. qRT-PCR Validation

The RNA-seq data were validated via RT-qPCR analyses using an Applied Biosystems StepOnePlus™ Real-Time System with the SYBR1 Select Master Mix (2×) (ABI, Los Angeles, CA, USA). The reaction protocol was as follows: 95 °C for 10 min; 45 cycles at 95 °C for 10 s, 60 °C for 30 s, and 72 °C for 20 s; and then 72 °C for 5 min. All the specific primers used are presented in Appendix A. Three technical replicates were included per sample. The rice *Ubiquitin* (*Os03g0234200*) gene was used as an internal standard. The relative expression values of the different genes were calculated using the 2^−ΔΔCt^ method.

## 5. Conclusions

In this study, we conducted an investigation into the transcriptome dynamics of the roots of two *indica* rice cultivars, SQSL and XZX45, under optimal temperature and cold stress conditions using RNA-seq technology. Our analysis focused on identifying differentially expressed genes (DEGs), the transcription factors associated with cold response, conducting GO analysis, KEGG pathway analysis, and analyzing the expression profiles of cold-response-related genes.

Through our comprehensive analysis, we discovered crucial DEGs and transcription factors that are specifically related to cold response. These findings provide insights into the molecular mechanisms underlying cold tolerance during the seedling stage in SQSL and XZX45. Notably, a significant proportion of the DEGs specific to SQSL were enriched in the “response to cold” gene ontology category (GO:0009409). This enrichment suggests that these DEGs likely play a critical role in the cold response mechanisms in rice.

Overall, our study contributes to a systematic understanding of the genetic and molecular basis of cold responses in rice. The knowledge gained from this study will serve as a foundation for future endeavors aimed at improving the cold tolerance of cold-sensitive rice varieties.

## Figures and Tables

**Figure 1 plants-12-02675-f001:**
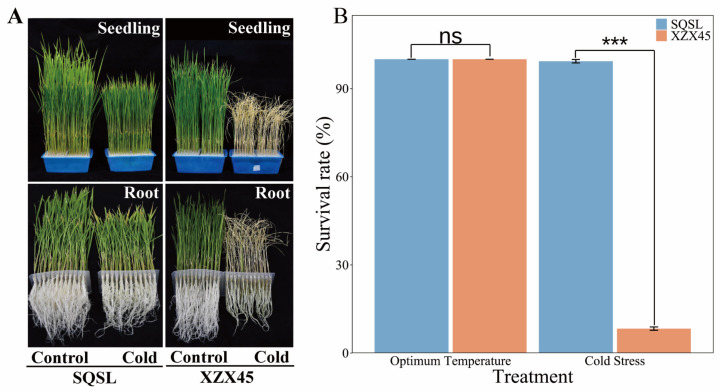
Comparison of the phenotype between SQSL and XZX45 under cold stress. (**A**) Morphological characteristics of SQSL and XZX45 seedlings (upper panel) and roots (lower panel) subjected to cold stress. (**B**) Survival rates of SQSL and XZX45 plants after a 5-day recovery period subsequent to a 5-day cold treatment (5 °C). Statistical significance is denoted as follows: ns (not significant) for *p* ≥ 0.05, and *** for *p* < 0.001.

**Figure 2 plants-12-02675-f002:**
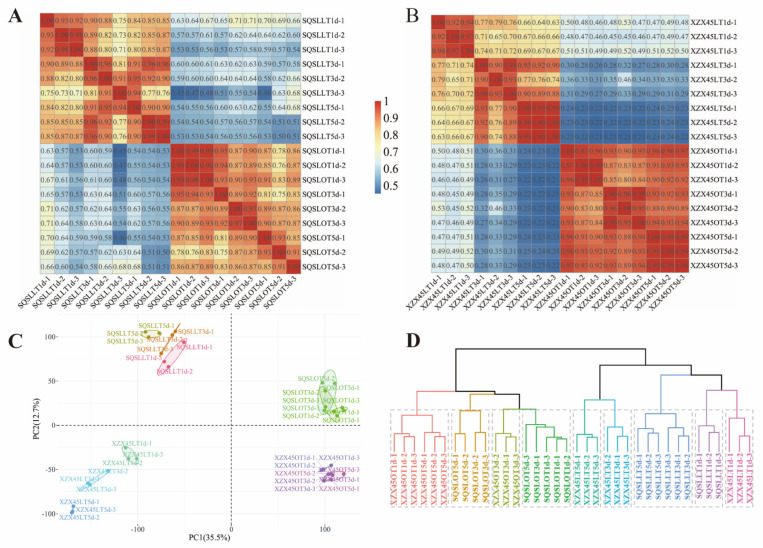
Transcriptome correlations during distinct stages of cold stress in SQSL and XZX45. (**A**) Pearson correlation coefficient (PCC) analysis of RNA-seq data obtained from three stages (1, 3, 5 days) of cold stress in SQSL. (**B**) Pearson correlation coefficient (PCC) analysis of RNA-seq data obtained from three stages (1, 3, 5 days) of cold stress in XZX45. (**C**) Principal component analysis (PCA) of samples at different stages of cold stress in SQSL and XZX45. Ellipses represent 95% confidence intervals. (**D**) Hierarchical cluster analysis of samples at different stages of cold stress in SQSL and XZX45.

**Figure 3 plants-12-02675-f003:**
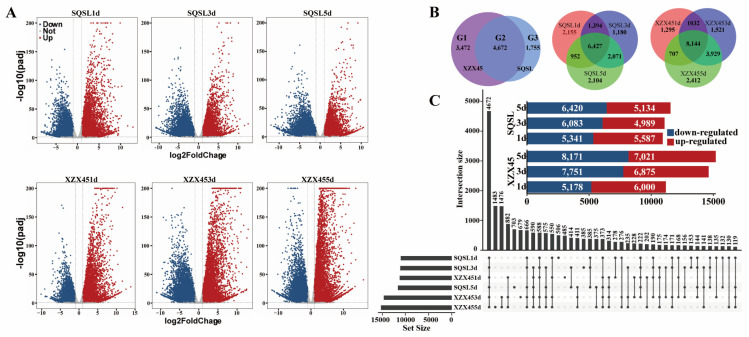
Statistical analysis of differentially expressed genes (DEGs) at distinct stages of cold stress in SQSL and XZX45. (**A**) Volcano plots illustrate the relationship between the false discovery rate (FDR) and fold change, displaying the DEGs identified in SQSL (upper panel) and XZX45 (lower panel) at different stages of cold stress. SQSL1d, SQSL3d, and SQSL5d represent 1, 3, and 5 days of cold treatment in SQSL, respectively, while XZX451d, XZX453d, and XZX455d represent the corresponding time points in XZX45. (**B**) Venn diagram demonstrating the overlap of DEGs at different stages of cold stress in SQSL and XZX45. Specifically, 3472 DEGs (labeled as G1) were specific to XZX45, while G2 represents the DEGs common to both SQSL and XZX45, and G3 represents the DEGs specific to SQSL. (**C**) Comparative analysis of DEGs at different stages of cold stress in SQSL and XZX45.

**Figure 4 plants-12-02675-f004:**
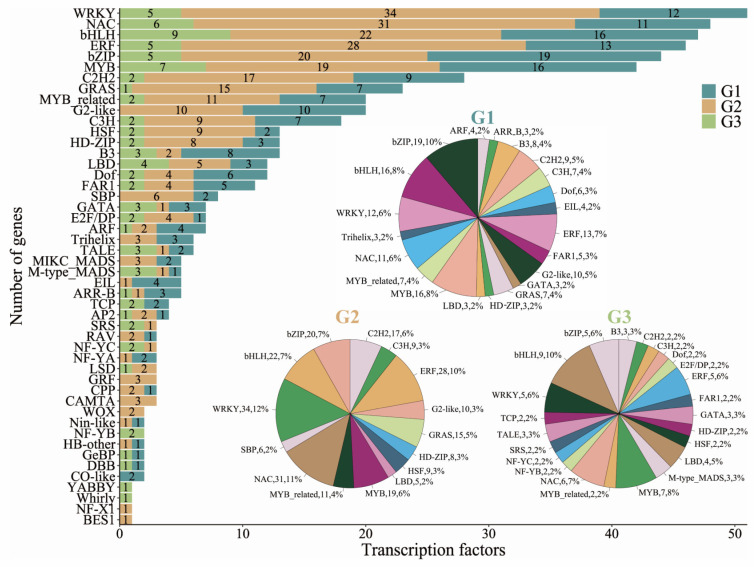
Transcription factors (TFs) analysis of differentially expressed genes (DEGs) identified in G1, G2, and G3. Pie chart illustrating the distribution of TF families, represented by their respective number and percentage, among all DEGs. Bar plot displaying the number of genes associated with different transcription factors in G1, G2, and G3.

**Figure 5 plants-12-02675-f005:**
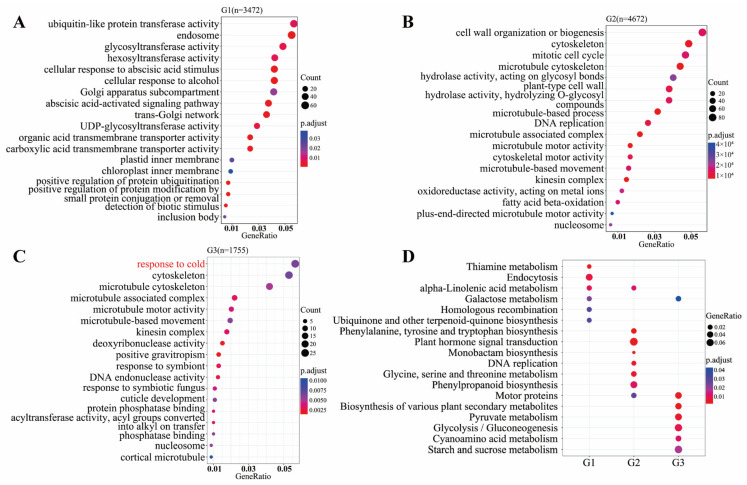
Functional annotation of differentially expressed genes (DEGs) in G1, G2, and G3. (**A**–**C**) Gene Ontology (GO) annotation of DEGs in G1, G2, and G3. The vertical axis represents the functional annotation information, while the horizontal axis represents the gene ratio of DEGs annotated to each specific function. (**D**) KEGG pathway enrichment comparison among G1, G2, and G3. The *p.adjust* value is represented by the color of the dot, where a smaller *p.adjust* value is indicated by a color closer to red. The size of the dots reflects the relative number of DEGs associated with each specific pathway.

**Figure 6 plants-12-02675-f006:**
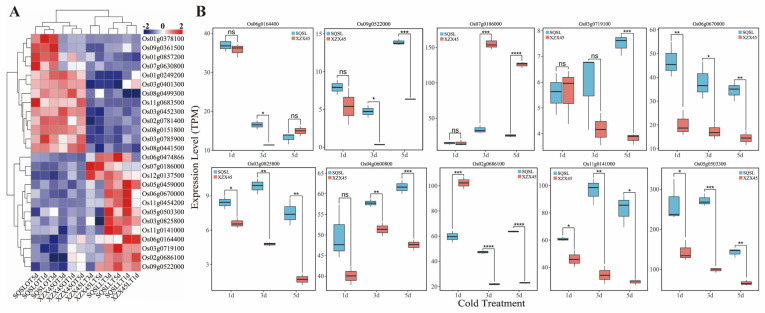
Expression profile of differentially expressed genes (DEGs) enriched in the “response to cold” Gene Ontology term (GO:0009409). (**A**) Heatmap depicting the expression levels of the DEGs enriched in the “response to cold” term. The heatmap is constructed based on TPM (transcripts per million) values across all samples. The color spectrum ranging from blue to red represents the gene expression levels from low to high. (**B**) Comparative analysis of the expression levels of selected DEGs between SQSL and XZX45 at different stages of cold stress. The data are presented as means ± SEM (standard error of the mean) from three independent replicates. Statistical significance is denoted as follows: ns (not significant) for *p* ≥ 0.05, * for *p* < 0.05, ** for *p* < 0.01, *** for *p* < 0.001, and **** for *p* < 0.0001.

**Table 1 plants-12-02675-t001:** List of differentially expressed genes (DEGs) enriched in the term “response to cold” (GO:0009409).

Gene IDs	Gene Symbol	Description	Log2foldchange					
			SQSL1d	SQSL3d	SQSL5d	XZX451d	XZX453d	XZX455d
*Os04g0600800*	*OsCOLD1*	Abscisic acid G-protein coupled receptor	0.78	0.85	1	0.58	1.1	0.86
*Os01g0249200*	*OsGLP1-1*	Cupin domain	−1.42	−2.64	−1.68	−0.12	−1.7	−1.07
*Os01g0378100*	*-*	Peroxidase family	−3.85	−6.52	−4.81	-	-	−0.8
*Os01g0857200*	*-*	UPP synthase family	−3.17	−2.33	−3.12	−2.83	0.66	−0.46
*Os02g0686100*	*OsATL32*	E3 ubiquitin ligase	1.93	2.92	2.15	1.98	2	0.31
*Os02g0781400*	*-*	GroES chaperonin family	−2.4	−2.69	−1.82	−0.87	−3.77	−2.25
*Os03g0401300*	*OsScS2*	Sucrose-UDP glucosyltransferase 2	−1.11	−4.84	−4.98	0.01	−5.09	−4.16
*Os03g0452300*	*-*	Ribosomal protein S5	−2.47	−1.81	−1.8	−1.47	−3.43	−2.93
*Os03g0719100*	*OsSIZ2*	SUMO E3-ligase	1.11	1.05	1.25	0.93	0.9	0.68
*Os03g0785900*	*OsGSTU1*	Glutathione transferase U1	−2.1	−3.36	−4.88	−1.58	−4.83	−4.04
*Os03g0825800*	*OsRLCK120*	Receptor-like cytoplasmic kinase 120	1.77	1.59	1.26	1.45	1.44	0.01
*Os05g0459000*	*OsDLN143*	Myb-like DNA-binding domain	1.62	1.29	1.12	1.92	1.09	0.74
*Os05g0503300*	*OsSiR*	Nitrite/Sulfite reductase ferredoxin-like half domain	1.6	1.33	1.41	1.33	1.06	0.48
*Os06g0164400*	*OsbHLH108*	Helix-loop-helix DNA-binding domain	2.64	1.73	1.14	2.34	1.37	0.74
*Os06g0474866*	*-*	Similar to Alpha-glucan water dikinase	2.65	3.72	4.22	1.02	3.26	5.57
*Os06g0670000*	*OsABA3*	Molybdenum cofactor sulfurase, ABA deficient 3	2.03	1.85	1.29	1.02	1.33	0.97
*Os07g0186000*	*OsTRXh1*	Thioredoxin family	1.6	2.68	2.13	1	4.97	4.88
*Os07g0630800*	*-*	Lactate/malate dehydrogenase	−5.77	−6.8	−6.01	−4.56	−1.4	−0.16
*Os08g0151800*	*OsMDAR5*	Pyridine nucleotide-disulfide oxidoreductase	−1.52	−1.62	−1.57	−0.75	−1.34	−0.93
*Os08g0441500*	*OsCCR20*	Cinnamoyl-CoA reductase	−2.11	−1.34	−2.27	−2.36	−0.56	−2.8
*Os08g0499300*	*OsWRKY30*	WRKY transcription factor 30	−3.85	−2.07	−1.11	−4.31	−1.39	0.34
*Os09g0361500*	*OsICS1*	Chorismate binding enzyme	−1.56	−3.51	−2.74	0.16	−0.81	−1.04
*Os09g0522000*	*OsDREB1B*	Dehydration-responsive element-binding protein 1B	3.15	4.28	5.65	2.52	2.42	1.94
*Os11g0141000*	*-*	TAP42-like family	1.03	1.55	1.48	1.1	1.31	0.85
*Os11g0454200*	*OsRAB16B*	Responsive to ABA gene 16B	9.01	4.45	5.13	5.76	5.55	4.5
*Os11g0683500*	*OsBGLU36*	Glycosyl hydrolase 1 family	−4.55	−5.04	−5.92	−5.04	−4.2	−2.27
*Os12g0137500*	*-*	TAP42-like family	1.1	1.63	1.43	0.94	2.2	1.75

## Data Availability

All raw sequencing reads generated in this study have been deposited in the public database of the National Genomics Data Center under the data accession number PRJCA017633. All the other data are available from the corresponding authors upon request.

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
