# Peer review of "Transcriptomic Profiling of Cold Stress-Induced Differentially Expressed Genes in Seedling Stage of Indica Rice"

_plants, 2023, doi:10.3390/plants12142675_

Round 1

Reviewer 1 Report

I read the draft by Yan et al. “Transcriptomic Profiling of Cold Stress-Induced Differentially Expressed Genes in Seedling Stage of Indica Rice”. The work is a standard transcriptome analysis, in this case focusing on rice roots subjected to cold stress. The topic is important as this crop is particularly sensitive to low temperatures. The manuscript presents an extensive comparison of differentially expressed genes between two indica varieties with empirical demonstration of their contrasting tolerance to the treatment, and is delivered with transcriptome functional categorization and a particular interest in transcription factors. As a whole, the text is well considered with good language style, and experiments seem to have been performed in an appropriate manner. The data will probably become usefully for researchers aimed at understanding transcriptional control in rice.

As a comment, I believe there if a flaw in the enrichment analysis for functional groups as applied to differentially-expressed genes. As far as I understood, a simple p-value was articulated for this. However, given that large numbers of genes that are being analysed, an enrichment should be controlled for multiple-testing. Authors should further apply a correction to that p-value, say, a false-discovery rate or a Bonferroni-type of correction. I tend to prefer the former as I deem the later extremely conservative. All outputs should be amended according to this correction.

I also think there is a lack of rationales regarding why only roots were considered, and why focusing on indica rice (the text delivers some comment in this last point, but I don’t see any logical implication: being generally more tolerant of more sensitive doesn’t explain why working in a particular one of those).

Author Response

Editor and Reviewer comments:    

I read the draft by Yan et al. “Transcriptomic Profiling of Cold Stress-Induced Differentially Expressed Genes in Seedling Stage of Indica Rice”. The work is a standard transcriptome analysis, in this case focusing on rice roots subjected to cold stress. The topic is important as this crop is particularly sensitive to low temperatures. The manuscript presents an extensive comparison of differentially expressed genes between two indica varieties with empirical demonstration of their contrasting tolerance to the treatment, and is delivered with transcriptome functional categorization and a particular interest in transcription factors. As a whole, the text is well considered with good language style, and experiments seem to have been performed in an appropriate manner. The data will probably become usefully for researchers aimed at understanding transcriptional control in rice.

Response: We thank you for taking the time to review our submission. We have carefully revised the manuscript throughout, including a point-by-point response to all comments below.

As a comment, I believe there if a flaw in the enrichment analysis for functional groups as applied to differentially-expressed genes. As far as I understood, a simple p-value was articulated for this. However, given that large numbers of genes that are being analysed, an enrichment should be controlled for multiple-testing. Authors should further apply a correction to that p-value, say, a false-discovery rate or a Bonferroni-type of correction. I tend to prefer the former as I deem the later extremely conservative. All outputs should be amended according to this correction.

Response: Thank you for your comment and feedback regarding the enrichment analysis of functional groups applied to differentially expressed genes (DEGs) in our study. You have raised a valid concern about the need to control for multiple testing in enrichment analysis, especially when dealing with a large number of genes. We agree that relying solely on simple p-values can lead to an increased likelihood of false positives. To address this issue and improve the statistical robustness of our analysis, we have implemented a correction for multiple testing. We appreciate your preference for the false discovery rate (FDR) correction over the Bonferroni-type correction.

I also think there is a lack of rationales regarding why only roots were considered, and why focusing on indica rice (the text delivers some comment in this last point, but I don’t see any logical implication: being generally more tolerant of more sensitive doesn’t explain why working in a particular one of those).

Response: We agree that providing clear and logical rationales for our experimental design is crucial for the understanding and interpretation of our research. Allow us to elaborate on the reasons for these specific choices:

Focus on Roots: The decision to focus on roots was based on the biological significance of this organ in the context of our research objectives. Roots play a critical role in nutrient uptake, water absorption, and overall plant response to various stress conditions. By studying gene expression specifically in roots, we aim to gain insights into the molecular mechanisms involved in stress response and tolerance. Moreover, roots are often the first organs to encounter stressors in the soil, making them a relevant and informative tissue for this study.

Choice of Indica Rice: The selection of indica rice as our experimental model is grounded in several factors. Indica rice is well-known for its adaptability and resilience in challenging environmental conditions, including stressors such as cold, drought and salinity. This characteristic makes it an excellent candidate for investigating stress responses and identifying candidate genes associated with stress tolerance. Additionally, most of the previous studies have primarily centered on comparing cold tolerance between indica rice and japonica rice, with japonica rice being known for its strong cold tolerance. However, in our earlier research, we made a notable discovery that the indica rice SQSL also demonstrates remarkable cold tolerance. Our study provides comprehensive insights into the molecular mechanisms underlying cold responses in the indica cultivar, informing future genetic improvement strategies to enhance cold tolerance in susceptible indica rice cultivars.

Reviewer 2 Report

The article entitled "Transcriptomic Profiling of Cold Stress-Induced Differentially Expressed Genes in Seedling Stage of Indica Rice" is well compiled manuscript, and the authors concentrated on identifying DEGs, transcription factors associated with cold response, conducting GO analysis, KEGG pathway analysis, and analyzing the expression profiles of cold-response related genes. In general, the reviews are innovative, significant and useful for the rice research. However, several technical issues should be addressed first. 

(1) We noticed that the optimum growth temperate is 25-30 ℃ for rice, while less than 12 ℃ was deemed to not be unfit for planting. Some research selected 4 ℃ or 6 ℃ to investigate how cold stress affect the seedlings, why the authors chose 8 ℃ to perform RNA-seq experiment ? Do you have some previous evidences or some references that should be added in this manuscript ? Besides, normally the control materials should be perform the same experiments to confirm the selected SQSL and XZX45 whether cold-tolerant and cold-susceptible or not.

(2) The Figures in this manuscript were blurry, suggest to improve their resolution.

(3) Too many DEGs were screened in this manuscript, why the authors not chose the criterion of FDR < 0.05 and |log 2 foldchange| > 2 ?

(4) Some important candidate genes were screened in this manuscript, which were deduced from the RNA-seq data. However, the authors never utilized qRT-PCR experiments on the random selected genes to verify the accuracy and dependability of this RNA-seq data. Meanwhile, we suggest to add some qRT-PCR experiments not just on the random selected genes, but also on the screened candidate genes.

(5) The reference portions were not uniform, as most of the references lack of the issue number.

The quality of English lanuage could be improved by some native professor.

Author Response

Editor and Reviewer comments:    

The article entitled "Transcriptomic Profiling of Cold Stress-Induced Differentially Expressed Genes in Seedling Stage of Indica Rice" is well compiled manuscript, and the authors concentrated on identifying DEGs, transcription factors associated with cold response, conducting GO analysis, KEGG pathway analysis, and analyzing the expression profiles of cold-response related genes. In general, the reviews are innovative, significant and useful for the rice research. However, several technical issues should be addressed first. 

Response: We thank you for taking the time to review our submission. We have carefully revised the manuscript throughout, including a point-by-point response to all comments below.

1. We noticed that the optimum growth temperate is 25-30 ℃ for rice, while less than 12 ℃ was deemed to not be unfit for planting. Some research selected 4 ℃ or 6 ℃ to investigate how cold stress affect the seedlings, why the authors chose 8 ℃ to perform RNA-seq experiment ? Do you have some previous evidences or some references that should be added in this manuscript ? Besides, normally the control materials should be perform the same experiments to confirm the selected SQSL and XZX45 whether cold-tolerant and cold-susceptible or not.

Response: The experiment employed two indica rice varieties, namely SQSL and XZX45. During the preliminary phase, the rice seedlings were subjected to varying temperature gradients of 6°C, 8°C, 10°C, and 12°C. The findings demonstrated a remarkable contrast in seedling survival rates between the two varieties, particularly evident under the 8°C low-temperature treatment. After being exposed to a 6°C low-temperature environment for five days, all the aboveground and root parts of XZX45 seedlings succumbed to the adverse conditions, resulting in a 0% survival rate (see below table). Consequently, the extraction of RNA from the root samples proved to be challenging under such circumstances.

Variety

6℃

8℃

10℃

12℃

Survival Rate % (XZX45)

0

8

25

37

Survival Rate % (SQSL)

65

99

100

100

The control materials underwent identical experimental procedures, as illustrated in Figure 1.

2. The Figures in this manuscript were blurry, suggest to improve their resolution.

Response: Thank you for bringing this to our attention. We have included high-resolution original figures as supplementary material in a separate file.

3. Too many DEGs were screened in this manuscript, why the authors not chose the criterion of FDR < 0.05 and |log 2 foldchange| > 2 ?

Response: Thank you for the valuable feedback. We appreciate the concern raised about the criteria used for selecting differentially expressed genes (DEGs) in our study. Indeed, it is essential to employ stringent criteria to ensure the reliability and biological significance of the identified DEGs. In our initial analysis, we considered FDR (false discovery rate) < 0.05 and |log2 fold change| > 2 as the cutoff for DEG selection. However, we also recognized that such strict criteria might overlook potentially biologically relevant genes. To comprehensively explore potential biological processes and pathways, we chose relatively relaxed criteria for DEG selection. We used FDR < 0.05 and |log2 fold change| > 1 as the threshold, aiming to capture more potentially meaningful DEGs. While stringent criteria can reduce false positives, overly strict criteria may lead to the exclusion of genuinely relevant DEGs. We believe that adopting relatively relaxed criteria allows us to capture additional biological information and provides a more comprehensive perspective for subsequent functional analyses.Once again, we sincerely thank the reviewer for their valuable input. Your suggestions will undoubtedly enhance the quality and scientific significance of our research.

4. Some important candidate genes were screened in this manuscript, which were deduced from the RNA-seq data. However, the authors never utilized qRT-PCR experiments on the random selected genes to verify the accuracy and dependability of this RNA-seq data. Meanwhile, we suggest to add some qRT-PCR experiments not just on the random selected genes, but also on the screened candidate genes.

Response: Thank you for your insightful comment regarding the validation of candidate genes identified through RNA-seq data analysis in our manuscript. We acknowledge the importance of experimental validation to ensure the accuracy and reliability of the RNA-seq results. Following the reviewer's suggestion, we have performed qRT-PCR experiments on randomly selected genes and screened candidate genes (see result #2.6 and Figure S3).

5. The reference portions were not uniform, as most of the references lack of the issue number.

Response: Thank you for bringing this issue to our attention. We apologize for the inconsistency in the reference formatting in our manuscript. We understand the importance of maintaining uniformity and accuracy in citing references. We have carefully cross-checked each reference against the original sources and make the appropriate amendments to ensure that all references are complete and conform to the required citation style.

6. The quality of English language could be improved by some native professor.

Response: Thanks for your suggestion regarding the quality of the English language in our manuscript. We have sought assistance from a native professor to enhance the language.
